# Machine learning and dengue forecasting: Comparing random forests and artificial neural networks for predicting dengue burden at national and sub-national scales in Colombia

**Naizhuo Zhao**[1,2], **Katia Charland**[3], **Mabel Carabali**[4], **Elaine O. Nsoesie**[5], **Mathieu Maheu-Giroux**[4,6], **Erin Rees**[7], **Mengru Yuan**[4], **Cesar Garcia Balaguera**[8], **Gloria Jaramillo Ramirez**[8], **Kate Zinszer**[3,6,9]*

1 Department of Land Resource Management, School of Humanities and Law, Northeastern University, Shenyang, Liaoning, China, 2 Division of Clinical Epidemiology, McGill University Health Centre, Montreal, Quebec, Canada, 3 Centre for Public Health Research, Montreal, Quebec, Canada, 4 Department of Epidemiology, Biostatistics, and Occupational Health, School of Population and Global Health, McGill University, Montreal, Quebec, Canada, 5 Department of Global Health, Boston University, Boston, Massachusetts, United States of America, 6 Quebec Population Health Research Network, Montreal, Quebec, Canada, 7 Public Health Risk Sciences Division, National Microbiology Laboratory, Public Health Agency of Canada, Saint-Hyacinthe, Quebec, Canada, 8 Faculty of Medicine, Universidad Cooperativa de Colombia, Villavicencio, Meta, Colombia, 9 Department of Preventive and Social Medicine, School of Public Health, University of Montreal, Montreal, Quebec, Canada

* kate.zinszer@umontreal.ca

**Data Availability Statement:** The epidemiological data are freely available through www.ins.gov.co,

## Abstract

The robust estimate and forecast capability of random forests (RF) has been widely recognized, however this ensemble machine learning method has not been widely used in mosquito-borne disease forecasting. In this study, two sets of RF models were developed at the national (pooled department-level data) and department level in Colombia to predict weekly dengue cases for 12-weeks ahead. A pooled national model based on artificial neural networks (ANN) was also developed and used as a comparator to the RF models. The various predictors included historic dengue cases, satellite-derived estimates for vegetation, precipitation, and air temperature, as well as population counts, income inequality, and education. Our RF model trained on the pooled national data was more accurate for department-specific weekly dengue cases estimation compared to a local model trained only on the department's data. Additionally, the forecast errors of the national RF model were smaller to those of the national pooled ANN model and were increased with the forecast horizon increasing from one-week-ahead (mean absolute error, MAE: 9.32) to 12-weeks ahead (MAE: 24.56). There was considerable variation in the relative importance of predictors dependent on forecast horizon. The environmental and meteorological predictors were relatively important for short-term dengue forecast horizons while socio-demographic predictors were relevant for longer-term forecast horizons. This study demonstrates the potential of RF in dengue forecasting with a feasible approach of using a national pooled model to forecast at finer spatial

the sociodemographic data are freely available through www.dane.gov.co, and the environmental data are freely available through lpdaac.usgs.gov (MODIS products) and www.cpc.ncep.noaa.gov (CMORPH product).

**Funding:** This work was supported by seed grant funding provided by the Quebec Population Health Research Network to KZ and MMG, and by a grant from the Canadian Institutes of Health Research (428107) to KZ. The funders had no role in study design, data collection and analysis, decision to publish, or preparation of the manuscript.

**Competing interests:** The authors have declared that no competing interests exist.

scales. Furthermore, including sociodemographic predictors is likely to be helpful in capturing longer-term dengue trends.

## Author summary

Dengue virus has the highest disease burden of all mosquito-borne viral diseases, infecting 390 million people annually in 128 countries. Forecasting is an important warning mechanism that can help with proactive planning and response for clinical and public health services. In this study, we compare two different machine learning approaches to dengue forecasting: random forest (RF) and artificial neural networks (ANN). National (pooling across all departments) and local (department-specific) models were compared and used to predict future dengue cases in Colombia. In Colombia, the departments are administrative divisions formed by a grouping of municipalities. The results demonstrated that the counts of future dengue cases were more accurately estimated by RF than by ANN. It was also shown that environmental and meteorological predictors were more important for forecast accuracy for shorter-term forecasts while socio-demographic predictors were more important for longer-term forecasts. Finally, the national pooled model applied to local data was more accurate in dengue forecasting compared to the department-specific model. This research contributes to the field of disease forecasting and highlights different considerations for future forecasting studies.

## Introduction

Dengue virus is most prevalent of the mosquito-borne viral diseases, infecting 390 million people annually in 128 countries with four different virus serotypes [1]. Rising incidence and large-scale outbreaks are largely due to inadequate living conditions, naïve populations, global trade and population mobility, climate change, and the adaptive nature of the principal mosquito vectors *Aedes aegypti* and *Aedes albopictus* [2, 3]. The direct and indirect costs of dengue are substantial and impose enormous burdens on low- and middle-income tropical countries, with a global estimate of US$8.9 billion in costs per year [4].

Human and financial costs of dengue can be alleviated when response systems, such as intervention strategies, health care services, and supply chain management, receive timely warnings of future cases through forecasting models [5]. A number of dengue forecasting models have been developed and these models can be generally classified into two methodological categories: time series and machine learning [6, 7]. The majority of existing dengue forecasting models used time series methods and typically Autoregressive Integrated Moving Average (ARIMA), in which lagged meteorological factors (e.g. temperature and precipitation) act as covariates in conjunction with historical dengue data for one- to 12-week-ahead forecasting [8–13]. Many studies reported that conventional time series models such as ARIMA are insufficient to meet complex forecasting requirements [14–16], as multiple trends and outliers present in the time series reduce the forecasting accuracy [17].

In the last two decades, machine learning (ML) methods have been used in many disciplines, such as geography, environment, and epidemiology, to yield meaningful findings from highly heterogeneous data. Differing from statistical modeling that forms relationships between variables based on many assumptions (e.g. independence of predictor variables, homoscedasticity, and normal distributions of errors), machine learning facilitates the

inclusion of a large number of correlated variables, enable the modeling of complex interactions between variables, and can fit complex models without presupposing forms (e.g. linear, exponential, and logistic) of functions, providing a more flexible approach for disease forecasting [18, 19]. Decision trees, support vector machine, shallow neural network, K-nearest neighbor, gradient boosting, and naive Bayes are frequently used ML approaches in dengue-forecasting studies [7, 20–23]. Compared to the above ML methods, random forests (RF), another common ML algorithm, have shown to be more accurate in forecasting given its ability to overcome the common problem of over-fitting through the use of bootstrap aggregation [24–28].

Random forests have been used to forecast dengue risk in several countries including Costa Rica [29], Philippines [30, 31], Pakistan [32], Peru and Puerto Rico [33]. However, time or seasonal variables were not always included in the models nor were sociodemographic predictors, which have been found to improve forecast accuracy in HIV [34] and Ebola [35] epidemic models. Furthermore, dengue models, regardless of the use of the time series or ML approaches, have been developed for predicting dengue cases in individual administrative areas such in a city or a province [9–12, 20–23]. Universal dengue prediction models that are effective across different administrative regions remain scarce.

Historically, Colombia is one of the countries most affected by dengue, with the *Aedes* mosquitoes being widely distributed throughout all departments at elevations below 2,000 meters [36, 37]. The objective of this study was to evaluate the potential of RF forecasting models at the department and national level in Colombia. We compared the accuracy of department-specific RF models to a nationally-pooled RF model to understand the feasibility of using a pooled model to predict dengue cases for individual departments. Using ARIMA as baseline, we also compared errors of the nationally pooled RF model with those of Artificial Neural Network (ANN), another classic and widely used ML approach. Finally, we estimated the change in importance of different predictors according to forecast horizon.

## Methods

### Ethics statement

Ethical approval was obtained from the Health Research Ethics Board from the University of Montreal (18-073-CERES-D).

**Data.** Various data were used to develop the forecasting models, which included: dengue cases from surveillance data, environmental indicators from remote sensing data, and sociodemographic indicators such as population, income inequity, and education coverage (Table 1). The dengue case surveillance data were extracted from an electronic platform, SIVIGILA, created by the Colombia national surveillance program and was available at the department level. The national surveillance program receives weekly reports from all public health facilities that provide services to cases of dengue. the dengue cases reported by SIVIGILA were a mixture of probable and laboratory confirmed cases without distinguishing between the two different case definitions. Laboratory confirmation for dengue is based on a positive result from antigen, antibody, or virus detection and/or isolation [38]. Probable cases are based on clinical diagnosis plus at least one serological positive immunoglobulin M test or an epidemiological link to a confirmed case within 14 days prior to symptom onset. Cases are typically reported within a week with severe cases usually being reported immediately.

Precipitation, air temperature, and land cover type have been shown to be three important determinants of *Aedes* mosquito abundance and are often used as predictors in dengue forecasting [9, 11, 21, 39]. In this study, precipitation data was obtained from the CMORPH (Climate Prediction Center morphing method) daily estimated precipitation dataset [40]. The

**Table 1. Summary of indicators and data sources.**

| Indicator | Source | Temporal granularity | Format |
|---|---|---|---|
| Dengue cases | SIVIGILA (national surveillance program of Colombia) | Weekly | Tabular |
| Rainfall | CMORPH precipitation data from NOAA's CPC | Daily | Gridded |
| EVI | MOD13C1 from NASA's LP DAAC | 16-day | Gridded |
| Temperature | MOD11C2 from NASA's LP DAAC | 8-day | Gridded |
| Population | Colombian National Administrative Department of Statistics | Yearly | Tabular |
| Gini Index | Colombian National Administrative Department of Statistics | Yearly | Tabular |
| Education coverage | Colombian National Administrative Department of Statistics | Yearly | Tabular |

CPC: Climate Prediction Center; LP DAAC: Land Processes Distributed Active Archive Center; NOAA: National Oceanic and Atmospheric Administration; EVI: enhanced vegetation index; CMORPH: Climate Prediction Center morphing method; NASA: National Aeronautics and Space Administration.

land surface temperatures were extracted from the MODIS Terra Land Surface Temperature 8-day image products (MOD11C2.006). Enhanced vegetation index (EVI) estimates were obtained from the MODIS Terra Vegetation Indices 16-Day image products (MOD13C1.006). Several studies have shown that socio-demographic factors may influence dengue transmission and incidence as significantly as environmental factors [41–43]. Education influences people's knowledge and behaviours towards infectious diseases, as people with higher education more likely to adopt behaviours to reduce risks of infection compared to individuals with lower education [44]. Income also affects risk of infectious diseases, with those from higher income brackets often being less exposed and consequently, less at-risk of infection compared to individuals with lower income [45]. Given this, we included population, education coverage, and the Gini Index (a measure of income inequity) as potential predictors, which were retrieved from the Colombian National Administrative Department of Statistics.

**Random forests.**   Random forests (RF) is an ensemble decision tree approach [46]. A decision tree is a simple representation for classification in which each internal node corresponds to a test on an attribute, each branch represents an outcome of a test, and each leaf (i.e. terminal node) holds a class label. Decision trees can also be used for regression when the target or outcome variable is continuous. Bootstrap aggregation, commonly known as bagging, is the most distinctive technique used in RF and bagging requires training each decision tree with a randomly selected subsample of the entire training datasets.

**Data preprocessing.**   To ensure a consistent temporal granularity with the outcome variable, the daily precipitation data were aggregated to a weekly frequency. The 8-day land surface temperature and the 16-day EVI data were resampled to a weekly frequency using a spline interpolation [47]. We assigned a given department the same population, Gini Index, and education coverage values for all weeks within the same calendar year.

Colombia has 32 departments and the archipelago of San Andrés, Providencia, and Santa Catalina (commonly known as *San Andrés y Providencia*) is a department consisting of two island groups and 775 km away from mainland Colombia. Due to the frequent cloud contamination over the small island areas, it was not possible to have high-quality MODIS images products for weekly temperature or EVI value estimation. Vaupés department had only 30 confirmed dengue cases during 2014 to 2018. Therefore, the departments of San Andrés y Providencia and Vaupés were excluded from this study and data from the other 30 departments were used to train our models.

Weekly dengue data from 2014–2017 was used to train the RF models and the data from 2018 was used to evaluate the models. To simulate 'real life' forecasting, we did not include the 2018 data for the socio-demographic variables given that they are only produced annually

whereas the remote sensing data are more readily available. Based on historical (2010–2017) time series data, double exponential smoothing with an additive trend was used to estimate the values for 2018. The specific exponential smoothing functions were determined by the optimal decay option in the "forecast" package for R software through minimizing the squared prediction errors.

**Development of RF, ANN, and ARIMA models.** We first developed RF models for each department (hereafter referred to as the local level). Let the "current" week be $k$ and the number of confirmed dengue cases be $y$. Referring to the RF streamflow forecasting model developed by Papacharalampous and Tyralis [48], we used the numbers of current and previous 11 weeks dengue cases (i.e. $y_k, y_{k-1}, \ldots, y_{k-10}, y_{k-11}$) of a department to predict one-week-ahead dengue cases (i.e. $y_{k+1}$) for each department. The current and previous 11 weeks of rainfall, land surface temperature, EVI, population, Gini Index, and education coverage were also included as predictors. These values were selected as previous studies demonstrated that the optimal lags of meteorological variables used for dengue forecasting are usually not larger than 12 weeks [49–54]. In addition, the ordinal number of the forecast week (1–52 for the year of 2015, 2016, 2017, and 2018 and 1–53 for 2014) as well as year (2014–2018) were treated as two predictor variables to account for seasonality and long-term changing trend of dengue occurrence [55,56].

We then developed a RF model at the national scale, which consisted of pooled the data across each department. To train a national-scale RF model for forecasting $n$-week-ahead dengue cases (where $n \leq 12$), we used the same predictor and target variables as those used in the local $n$-week-ahead forecasting models. The difference between the local and the national pooled models was that the local $n$-week-ahead models were trained using 209-$n$ (209 = 53+52 +52+52) samples while the national model was trained using 6270-30$n$ [i.e. (209-n) ×30] samples. Through 10-fold cross-validations, we found that the common settings for the number of variables randomly sampled as candidates at each split (i.e. the number of features divided by three) and the minimum size of terminal nodes (i.e. five) were also optimal to avoid over-fitting in our RF models [57]. The specific RF models were fitted by "randomForest" in the R statistical computing environment and set 1000 trees for an ensemble of trees (forest) [58]. We found that further increasing the number of trees did not markedly decrease out-of-bag mean square errors of the RF models but only increased computation time.

Artificial Neural Network (ANN) is considered a classic ML approach and to highlight the advantage of prediction accuracy of the RF models, we developed an ANN model at the national scale. The ANN was composed of one input layer, three hidden layers, and one output layer. The ANN model used ReLU as an activation function to solve the problem of a vanishing gradient and avoids over-fitting through setting "dropouts". Jointly considering prediction accuracy and computation time, we set "epoch" and "batch size" of the ANN models as 100 and 32 respectively. The ANN models had the same 53 predictor variables as the RF models, resulting in 53 neurons in the input layer and one neuron in the output layer. The number of neurons in the hidden layer was decreased layer by layer as the shape of an inverted pyramid. The specific number of neurons and value of dropout of a hidden layer were determined by iterative attempts until the mean absolute error (MAE) of the prediction could not be further reduced [59] (see Table 2).

Standard univariate ARIMA developed at the local scale was used as the baseline to compare with the RF and ANN models. The Hyndman-Khandakar algorithm was used for automatic ARIMA modeling [60]. This algorithm first determines the number of non-seasonal differences needed for stationarity (i.e. $d$ in ARIMA) using repeated Kwiatkowski-Phillips-Schmidt-Shin (KPSS) tests. Then, the number of autoregressive terms and the number of

lagged forecast errors (i.e. $p$ and $q$ in ARIMA respectively) by minimizing Akaike's Information Criterion (AIC).

**Model evaluation.** The MAEs of the ARIMA, RF, and ANN models were calculated for the 52 weeks in 2018 by the actual and the predicted numbers of dengue cases. The accuracy comparison was performed for the local (department) and national (pooled) scales. When the comparison for an $n$-week-ahead prediction was conducted at the national scale, the predicted numbers of dengue cases by the 30 local RF models were additively combined and compared with the actual national values to calculate one MAE. When the comparison was implemented at the local scale, the national RF model was applied to each one of the 30 departments and then the predicted values were compared with the actual numbers of dengue cases to compute 30 individual MAEs. To improve intuitive interpretation and facilitate comparisons of one model's predictive performance across different departments and forecasting horizons, we used the relative MAE (RMAE) to evaluate model accuracy [61]. We defined a *RMAE* between a ML (i.e. RF or ANN) and the baseline models at a horizon $h$ as:

$$RMAE_{A,B,h} = \frac{MAE_{A,h}}{MAE_{B,h}} \tag{1}$$

where $A$ represented a ML model and $B$ denoted the baseline ARIMA model.

Given that dengue burden varies across different years, we conducted leave-one-season-out cross-validations to improve the robustness of our evaluation. The accuracy between the national (pooled) and local RF models as well as the national ANN model were compared using RMAE. In the validations, four years of data were used to train the models and the remaining one year was used to validate the models. This procedure was iterated five times to ensure each year data were selected once for validation. An ARIMA model requires continuous time series and therefore, was not suitable for conducting the leave-one-season-out cross-validations. The specific ANN and ARIMA fitting processes were completed using the "keras" and "forecast" packages respectively in the R statistical computing environment.

Percentage of increased mean squared error (%IncMSE) is a robust and informative indicator to quantitatively evaluate the importance of predictor variables in a random forests model [62]. Percentage of increased mean squared error indicates the increase in the mean squared error (MSE) of prediction as a result of an independent variable being randomly shuffled while maintaining the other independent variables as unchanged [46]. A larger %IncMSE of a predictor variable suggests greater importance of the variable on the model's overall forecast accuracy and the %IncMSE was calculated for each predictor in each RF model.

## Results

An exceptionally large dengue outbreak occurred in Colombia during the study period. The counts of confirmed dengue cases reached more than 2,500 per week by the end of 2015 and the outbreak ended mid-year in 2016. Following this outbreak, the yearly dengue case peaks were drastically reduced in 2016 and 2017 but began increasing again in 2018 (Fig 1).

For any of the n-week-ahead (n≤12) forecasts, the national RF model more accurately predicted the counts of dengue cases than the ARIMA models, demonstrated by the smaller-than-

**Table 2. The numbers of neurons and values of dropouts in the hidden layers of the ANN models.**

| Hidden layer | Number of neurons | Dropout |
|:---:|:---:|:---:|
| First | 48 | 0.3 |
| Second | 32 | 0.2 |
| Third | 19 | 0.1 |

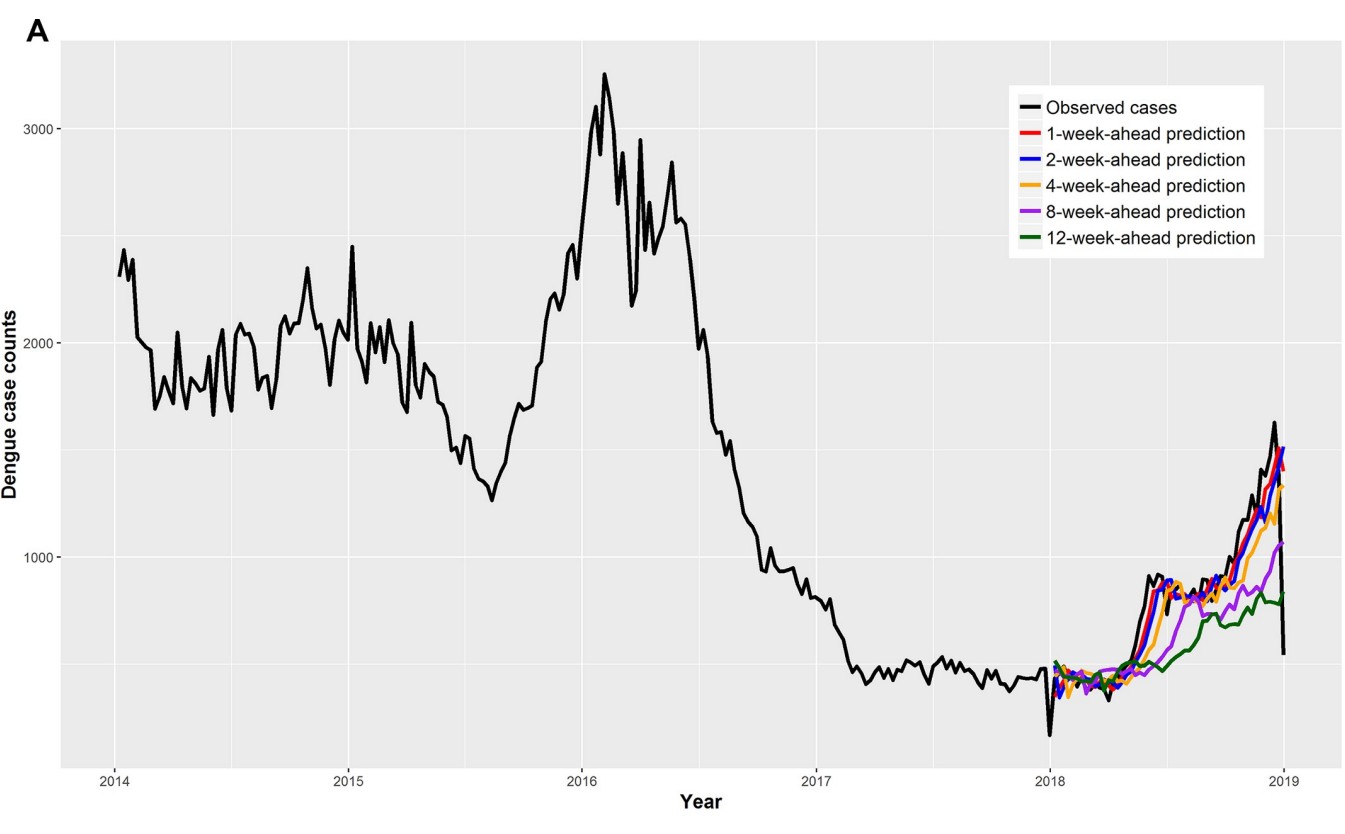

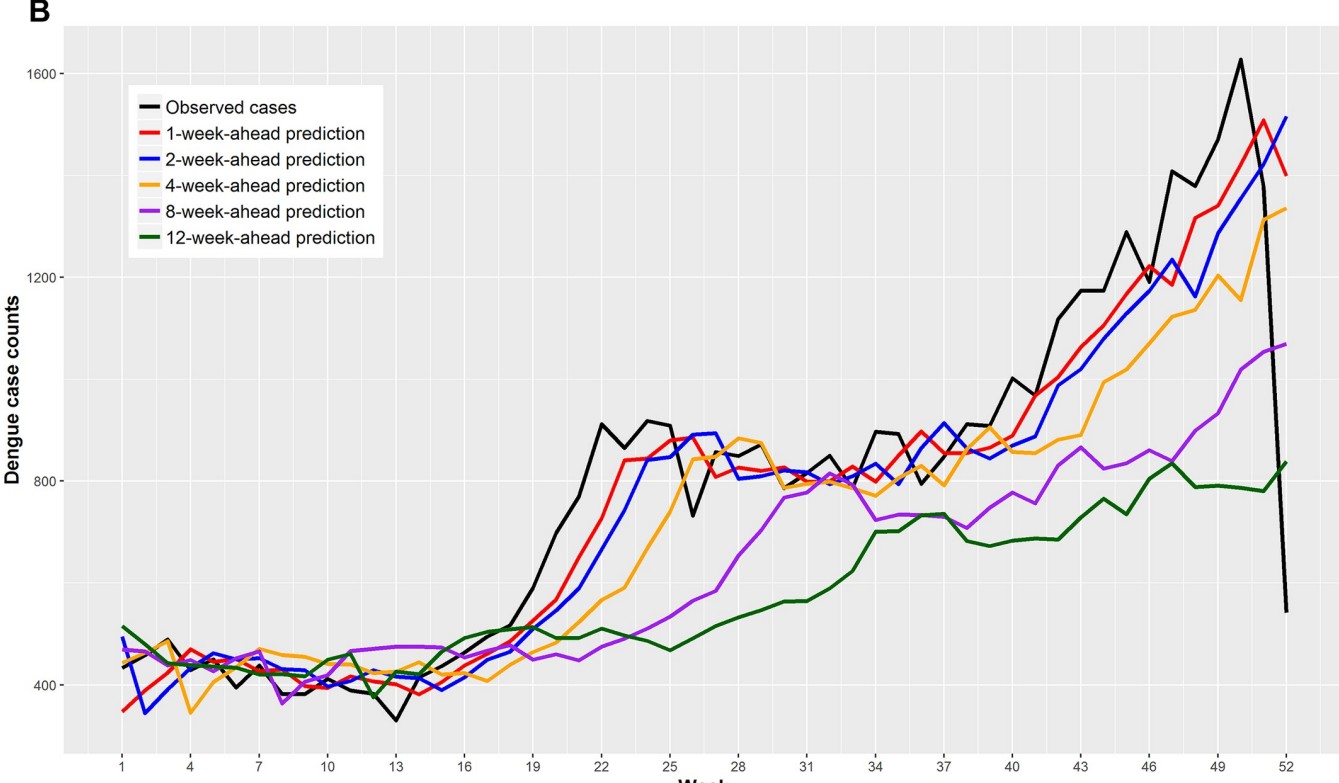

**Fig 1.** Weekly total counts of confirmed dengue cases over Colombia for 2014–2018 (A) and the predicted counts of dengue cases by the national one-, two-, four-, eight-, and twelve-week-ahead models for 2018 (B). See S1 Fig for the predicted counts of dengue cases by the remaining seven models.

**Table 3. Accuracy comparison among ARIMA, RF, and ANN model for prediction of 2018.**

| n-week ahead | MAE | RMAE | | |
|:---:|:---:|:---:|:---:|:---:|
| | ARIMA | Local RF | National RF | National ANN |
| 1 | 6.24 | 1.28 | 0.93 | 0.98 |
| 2 | 7.15 | 1.27 | 0.95 | 1.03 |
| 3 | 8.12 | 1.25 | 0.94 | 1.04 |
| 4 | 8.95 | 1.23 | 0.95 | 0.99 |
| 5 | 9.76 | 1.24 | 0.95 | 0.98 |
| 6 | 10.69 | 1.20 | 0.94 | 0.96 |
| 7 | 11.61 | 1.16 | 0.93 | 0.98 |
| 8 | 12.50 | 1.12 | 0.92 | 0.98 |
| 9 | 13.31 | 1.08 | 0.90 | 1.00 |
| 10 | 14.05 | 1.04 | 0.89 | 0.99 |
| 11 | 14.84 | 1.00 | 0.87 | 0.95 |
| 12 | 15.56 | 0.97 | 0.86 | 0.95 |

MAE: mean absolute error; RMAE: relative mean absolute error; ARIMA: Autoregressive Integrated Moving Average; RF: random forests; ANN: artificial neural network.

one RMAE (Table 3). The performance of the national model was better than that of the local model, demonstrated by the smaller overall RMAE and MAE (Tables 3 and 4). Moreover, in most cases, a department's dengue cases were more accurately predicted by the national model than the local model (Fig 2). The errors of the national RF model were mainly derived from under-estimation of cases which coincided with dramatic increases in cases towards the end of 2018. As expected, the under-estimation was more pronounced when predictions were made over a longer time period.

The overall MAE of the ANN model developed at the national scale and obtained from the leave-one-season-out cross-validation was smaller than that of the local RF model at any forecasting horizon (Table 4). The MAE grew for the ANN model with longer forecasting horizons compared to the local RF model. The RMAE of the ANN model obtained from the validation for 2018 was consistently smaller than that of the local RF model for each forecasting horizon.

**Table 4. Average MAEs of the leave-one-season-out cross-validations.**

| n-week ahead | Local RF | National RF | National ANN |
|:---:|:---:|:---:|:---:|
| 1 | 13.86 | 9.32 | 10.20 |
| 2 | 15.90 | 11.05 | 12.40 |
| 3 | 17.70 | 12.50 | 13.89 |
| 4 | 19.45 | 14.19 | 16.04 |
| 5 | 20.88 | 15.81 | 16.61 |
| 6 | 22.00 | 17.36 | 18.55 |
| 7 | 23.14 | 18.88 | 20.46 |
| 8 | 24.10 | 20.29 | 22.14 |
| 9 | 25.08 | 21.55 | 22.57 |
| 10 | 25.69 | 22.63 | 23.86 |
| 11 | 26.16 | 23.82 | 24.28 |
| 12 | 26.76 | 24.56 | 25.25 |

MAE: mean absolute error; RF: random forests; ANN: artificial neural network.

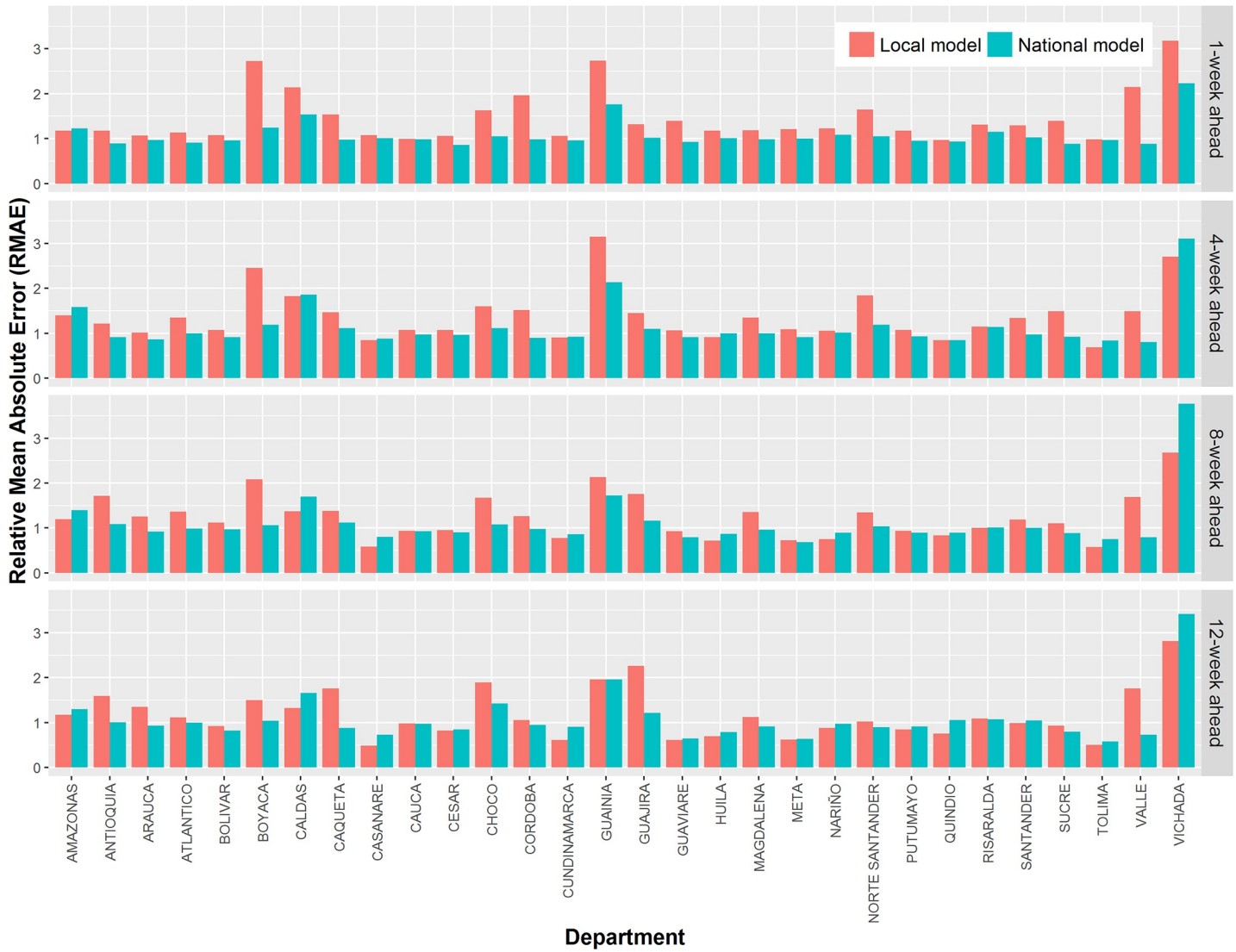

**Fig 2. Accuracy comparison between the local and the national random forests models at the department scale for the one-week ahead, four-week ahead, eight-week ahead, and twelve-week ahead predictions.** See S2 Fig for the comparison between the two types of models for all week ahead predictions.

The MAE and RMAE of the national RF model were always smaller than those of the national ANN model at any forecasting horizon.

The relative importance of different predictor variables in the national RF model was varied (Table 5). Firstly, "current" and "near current" past dengue data were extremely important in predicting occurrence of dengue in the near future (e.g. one- to three-weeks ahead). However, with the predicted week increasingly further away from the "current" week, the importance of historical dengue data decreased while the "current" week of dengue cases remained one of the top three most important predictors in predicting the future dengue cases. Secondly, the environmental (EVI) and the meteorological predictors (rainfall and temperature) were more important than the socio-demographic predictors when dengue cases were predicted in the near future (one- to three-weeks ahead). Yet, with the predicted week increasingly far away from the "current" week, importance of the three socio-demographic covariates (education, population, and Gini Index) became increasingly notable. Finally, the week predictor, which

**Table 5. The top ten most important predictor variables for predicting dengue cases in the national models, ordered from the largest to the smallest %IncMSEs.**

| Rank | 1 | 2 | 3 | 4 | 5 | 6 | 7 | 8 | 9 | 10 |
|---|---|---|---|---|---|---|---|---|---|---|
| 1-week-ahead | $Dengue_k$ (26.35%) | $Dengue_{k-1}$ (17.97%) | $Dengue_{k-2}$ (12.61%) | $Dengue_{k-3}$ (10.36%) | Week (8.78%) | $Dengue_{k-4}$ (7.83%) | $EVI_{k-11}$ (6.43%) | $Temperature_{k-11}$ (6.39%) | $EVI_{k-10}$ (6.07%) | $EVI_{k-8}$ (6.05%) |
| 2-week-ahead | $Dengue_k$ (25.72%) | $Dengue_{k-1}$ (17.13%) | Week (12.33%) | $Dengue_{k-2}$ (12.30%) | $Dengue_{k-3}$ (9.73%) | $Temperature_{k-11}$ (8.87%) | $Dengue_{k-4}$ (8.82%) | $EVI_{k-7}$ (8.42%) | $EVI_{k-5}$ (8.06%) | $EVI_{k-8}$ (7.41%) |
| 3-week-ahead | $Dengue_k$ (27.16%) | $Dengue_{k-1}$ (17.54%) | Week (14.57%) | $Dengue_{k-2}$ (12.91%) | $EVI_{k-8}$ (9.67%) | $EVI_{k-10}$ (8.52%) | $Temperature_{k-10}$ (8.49%) | Education (8.40%) | $Dengue_{k-3}$ (7.48%) | $Dengue_{k-4}$ (7.40%) |
| 4-week-ahead | $Dengue_k$ (27.24%) | Week (17.94%) | $Dengue_{k-1}$ (15.10%) | Education (12.97%) | $Dengue_{k-2}$ (11.28%) | $Temperature_{k-9}$ (10.03%) | $EVI_{k-8}$ (9.68%) | $Temperature_{k-11}$ (8.67%) | $EVI_{k-7}$ (8.37%) | $Dengue_{k-3}$ (7.86%) |
| 5-week-ahead | $Dengue_k$ (25.39%) | Week (18.86%) | $Dengue_{k-1}$ (18.73%) | Education (12.99%) | $Dengue_{k-2}$ (12.39%) | $EVI_{k-10}$ (11.42%) | $Temperature_{k-8}$ (11.15%) | $Temperature_k$ (11.31%) | Gini (10.33%) | $EVI_{k-9}$ (9.82%) |
| 6-week-ahead | $Dengue_k$ (24.88%) | Week (20.14%) | $Dengue_{k-1}$ (17.68%) | Education (17.13%) | Population (12.38%) | Year (11.83%) | $Dengue_{k-2}$ (11.54%) | $EVI_{k-8}$ (11.52%) | $EVI_{k-9}$ (11.24%) | $EVI_{k-1}$ (11.15%) |
| 7-week-ahead | $Dengue_k$ (25.61%) | Week (19.71%) | Education (17.66%) | $Dengue_{k-1}$ (17.49%) | Year (15.64%) | $Dengue_{k-2}$ (14.45%) | Population (12.49%) | Gini (11.69%) | $EVI_{k-10}$ (11.55%) | $EVI_{k-9}$ (11.06%) |
| 8-week-ahead | $Dengue_k$ (25.68%) | Week (21.49%) | Population (20.67%) | Education (19.16%) | $Dengue_{k-1}$ (16.84%) | Year (16.06%) | $Temperature_{k-11}$ (12.99%) | $Temperature_{k-5}$ (12.11%) | $Dengue_{k-2}$ (11.66%) | Gini (11.63%) |
| 9-week-ahead | $Dengue_k$ (24.11%) | Week (22.15%) | Population (21.56%) | Education (20.47%) | Year (17.70%) | $Dengue_{k-1}$ (17.44%) | $Temperature_{k-11}$ (12.94%) | $Dengue_{k-11}$ (12.05%) | Gini (11.89%) | $Temperature_{k-3}$ (11.15%) |
| 10-week-ahead | $Dengue_k$ (23.42%) | Week (23.03%) | Year (21.45%) | Education (20.38) | Population (19.80%) | $Dengue_{k-1}$ (17.22%) | Gini (14.88%) | $Dengue_{k-11}$ (13.02%) | $Temperature_{k-4}$ (12.95%) | $Dengue_{k-2}$ (10.60%) |
| 11-week-ahead | Year (22.94%) | Week (21.73%) | $Dengue_k$ (21.37%) | Population (18.61%) | Education (17.20%) | Gini (16.98%) | $Dengue_{k-1}$ (16.56%) | $Temperature_{k-11}$ (15.48%) | $Dengue_{k-10}$ (13.47%) | $Temperature_{k-4}$ (11.80%) |
| 12-week-ahead | Population (26.76%) | Year (24.86%) | $Dengue_k$ (22.50%) | Week (22.45%) | Education (17.12%) | Gini (17.72%) | $Dengue_{k-11}$ (16.71%) | $Dengue_{k-1}$ (16.67%) | $Dengue_{k-10}$ (14.06%) | $Temperature_{k-10}$ (13.07%) |

Dengue indicates historical dengue cases and EVI denotes enhanced vegetation index. %IncMSE: percentage of increased mean squared error.

accounted for the seasonal pattern of dengue, was important across all forecasting horizons but relatively smaller in importance with smaller forecasting horizons (i.e. $n \leq 4$).

## Discussion

In the current study, we developed a national pooled model to predict counts of dengue cases across different departments of Colombia and found that for the majority of departments, the national model more accurately forecasted future dengue cases at the department level compared to the local model. This result indicates the similarity in importance of dengue drivers across different administrative regions of Colombia. Random forests is an unsupervised tree-based regression approach requiring a relatively large training sample for the repeated splitting of the dataset into separate branches. A RF regression model cannot yield predictions for data points beyond the scope of the training data range. Pooling data from individual departments creates a training dataset with larger ranges of variables, increasing the extrapolating capacity of the RF model. Therefore, the national pooled model trained by a larger dataset had higher prediction accuracy compared to the local models. The national and the local models performed poorly in departments of Guainía and Vichada. The small population and consequently the low counts of dengue cases resulted in the relatively large errors in the two departments.

We also found that the meteorological and environmental variables were more important for prediction accuracy at smaller forecasting horizons compared to the socio-demographic variables, with socio-demographics being more important at larger forecasting horizons. This

is likely due to the influence of meteorological and environmental conditions on *Aedes* mosquitoes and the lag effects are usually between 1 to 4 weeks for temperature and precipitation [63–65]. Poor quality housing and sanitation management with high population density are key risk factors for dengue transmission [66, 67], and are closely related to education and poverty [68, 69]. These results demonstrate the complementary nature of these different groups of predictor variables and the importance of their inclusion in dengue forecasting models.

We compared our RF pooled national models to pooled national ANN models using the same predictor variables. Theoretically, with ANN, more complex correlations between predictor and target variables can be discerned by deeper (i.e. more hidden layers) networks [70]. However, traditional ANNs cannot handle the problem of vanishing gradient which results in the failure of improving accuracy of ANN models by adding more hidden layers. In the current study, we used the activation function of ReLU to overcome the issue of vanishing gradient, mitigated over-fitting by adding dropouts for each hidden layer, and predicted dengue cases with a three-hidden neural network. Compared with the ARIMA and local RF models, the ANN model developed by the national pooled data showed a stronger capability on forecasting dengue cases in Colombia across different forecasting horizons but performed slightly worse than the national RF model in this forecasting case study. It usually requires several iterative attempts to determine an optimal structure of an ANN model. By contrast, RF has conventional settings for tuning the hyperparameters (e.g. using the number of features divided by three for the number of variables at each split and five for the minimum size of terminal nodes) with the default hyperparameters having been found to be optimal in different studies [57].

Despite the strengths of our study, our RF approach is likely to generate time lags in forecasting rapid changes in dengue, which is also a common occurrence with other forecasting approaches. Including a predictor of mosquito abundance from an entomological surveillance program may reduce such time lag errors [71]. However, this type of data was not available at the national level given insufficient temporal and spatial granularity. Additionally, RF, as a non-parametric black-box approach, cannot use specific equations to quantify the relationships between the count of dengue cases and the heterogeneous predictor variables, although it is able to more flexibly and accurately capture the possibly complex non-linear and non-additive relationships among the variables. A more severe limitation of the RF model is the fact that RF cannot obtain values beyond the range of the variable in the training dataset. If an unprecedented dengue outbreak occurred in future, under-estimations will occur inevitably using the RF approach. Modeling changes in the count of dengue cases rather than the count may reduce such under-estimation errors.

Forecasting is an important warning mechanism that can help with proactive planning and response for clinical and public health services. This study highlights the potential of RF for dengue forecasting and also demonstrates the benefits of including socio-demographic predictors. Our findings also found that a national pooled model, on average, performed better compared to the local models. These findings have important implications for dengue forecasting models in public health in terms of time savings, such as pooled data versus locally-specific models, and predictors and approaches that could help improve forecast accuracy. Future studies should consider the inclusion of other arboviruses as predictors, such as chikungunya and Zika as well as examine the importance of other socio-economic factors. In addition, other promising ML methods should be tested including recurrent neural networks, which inherently account for time, and are able to capture complicated non-linear and non-additive relationships between predictor and target variables [72].

## Supporting information

**S1 Fig. Weekly total counts of confirmed dengue cases over Colombia for 2014–2018 and the predicted counts of dengue cases by the national three-, five-, six-, seven-, nine-, and eleven-week-ahead models for 2018.**
(TIFF)

**S2 Fig. Accuracy comparison between the local and the national random forests models at the department scale for each week ahead predictions using the relative mean absolute error (RMAE).**
(PDF)

## Author Contributions

**Conceptualization:** Naizhuo Zhao, Katia Charland, Elaine O. Nsoesie, Mathieu Maheu-Giroux, Erin Rees, Cesar Garcia Balaguera, Gloria Jaramillo Ramirez, Kate Zinszer.

**Data curation:** Mabel Carabali, Cesar Garcia Balaguera, Gloria Jaramillo Ramirez, Kate Zinszer.

**Formal analysis:** Naizhuo Zhao.

**Funding acquisition:** Mathieu Maheu-Giroux, Kate Zinszer.

**Investigation:** Naizhuo Zhao, Katia Charland, Mabel Carabali, Kate Zinszer.

**Methodology:** Naizhuo Zhao, Kate Zinszer.

**Project administration:** Naizhuo Zhao, Kate Zinszer.

**Resources:** Kate Zinszer.

**Software:** Naizhuo Zhao.

**Supervision:** Katia Charland, Elaine O. Nsoesie, Kate Zinszer.

**Validation:** Katia Charland, Cesar Garcia Balaguera, Gloria Jaramillo Ramirez.

**Visualization:** Mengru Yuan.

**Writing – original draft:** Naizhuo Zhao, Kate Zinszer.

**Writing – review & editing:** Naizhuo Zhao, Katia Charland, Mabel Carabali, Elaine O. Nsoesie, Mathieu Maheu-Giroux, Erin Rees, Mengru Yuan, Cesar Garcia Balaguera, Gloria Jaramillo Ramirez, Kate Zinszer.

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
