## [Decision Letter · Decision Letter 0]

3 Apr 2020

Dear Dr Zinszer,

Thank you very much for submitting your manuscript "Machine learning and dengue forecasting: Comparing random forests and artificial neural networks for predicting dengue burdens at the national sub-national scale in Colombia" for consideration at PLOS Neglected Tropical Diseases. As with all papers reviewed by the journal, your manuscript was reviewed by members of the editorial board and by several independent reviewers. In light of the reviews (below this email), we would like to invite the resubmission of a significantly-revised version that takes into account the reviewers' comments. 

We cannot make any decision about publication until we have seen the revised manuscript and your response to the reviewers' comments. Your revised manuscript is also likely to be sent to reviewers for further evaluation.

Sincerely,

Marc Choisy

Guest Editor

Robert Reiner

Deputy Editor

Reviewer's Responses to Questions

**Key Review Criteria Required for Acceptance?**

**Methods**

-Are the objectives of the study clearly articulated with a clear testable hypothesis stated?

-Is the study design appropriate to address the stated objectives?

-Is the population clearly described and appropriate for the hypothesis being tested?

-Is the sample size sufficient to ensure adequate power to address the hypothesis being tested?

-Were correct statistical analysis used to support conclusions?

-Are there concerns about ethical or regulatory requirements being met?

Reviewer #1: The clearly articulated objective of the study is "to evaluate the potential of using random forest forecasting models at the department and national levels in Columbia."

The study design needs an appropriate baseline model to truly address the stated objective. In the introduction, the authors state that "conventional time-series models such as ARIMA are insufficient to meet complex forecasting requirements", however that is only known because they are compared to baseline models that are commonly used by public officials. In a way, the authors use the Local Model as the baseline, but we have no reason to trust this as a baseline model because we would like to evaluate this model as well. For predictions of Dengue_{k+1}, a good naive baseline model would be Dengue_k because public health officials often react to the most recent data point. For longer term predictions, it would be good to show how the RFs perform relative to ARIMA models. A paper that describes selecting a baseline model for dengue prediction in detail is Reich et al., 2015, Case Study in Evaluating Time Series Prediction Models Using the Relative Mean Absolute Error.

An appropriate baseline model would allow the authors to calculate the relative mean absolute error which is a better measure of model performance than absolute measures such as MAE and RMSE.

Are there time lags in the dengue case data? For instance, dengue cases in Thailand had a substantial reporting lag (Reich et al., 2016. Challenges in real-time prediction of infectious disease: A case study of dengue in Thailand) and influenza cases in the US have about a 4-week reporting lag. If so, how does that affect the potential of using the RF forecasting model in real time?

I assume that the dengue cases are the sum of probable and confirmed dengue cases from SIVIGILA. Do you know how many cases were probable and confirmed for each week? I'd like to see a supplemental figure of both plus dengue hemorrhagic fever (or severe dengue), if possible.

Reviewer #2: The methods descriptions are sound and generally clear, but RF and ANN have plenty of details, and it would be nice to have more comprehensive algorithm descriptions. Particularly important are the issues of (a) how hyperparameter tuning was performed, and (b) whether regularization was used.

Reviewer #3: The present study combines climate and socio-economic data as well as past Dengue case count data to predict future Dengue cases in Colombia . The objective of using either subnational data separately or in combination for training for prediction is clearly stated. The choice of random forest algorithm is well motivated to avoid overfitting.

**Results**

-Does the analysis presented match the analysis plan?

-Are the results clearly and completely presented?

-Are the figures (Tables, Images) of sufficient quality for clarity?

Reviewer #1: The analysis presented matches the analysis plan and results are presented clearly.

Figure 1 is okay, but if a baseline model is chosen then I would prefer to see a relative MAE figure in place of figure 2. RMSE or MAE are absolute measures that might not be comparable across locations. If one location is more variable than another, that figure may merely be displaying the difference in variability. A figure of relative MAE would show the forecasting skill by location. The sentence in the Results describing Figure 2 (on lines 245-246) is accurate, however the assertion in the discussion (lines 309-310) may change if these locations are inherently more difficult to predict for a baseline model as well.

Is there a way to get the %IncMSEs into Table 3? The rank order is nice, but it doesn't tell us how much stronger the first variable is than the second, third, or tenth. If the lower impact variables have very low values, they could be left off in order to free up space for some numbers.

Reviewer #2: The results match the descriptions, and are clearly presented. However, there needs to be evaluation on more data, e.g., by using leave-one-season-out cross-validation. Some additional points of investigation noted in the comments might expand the contribution of this work and reinforce the points it makes.

Reviewer #3: The results are clearly presented in emphasizing that training on subnational data combined is superior to training separately to achieve high accuracy. The way how predictive power of different features depending on the time horizon chosen is interesting, e.g. socio-economic factors seem to be more important for long-range predictions.

Figure 1 is hardly legible, please increase the resolution.

**Conclusions**

-Are the conclusions supported by the data presented?

-Are the limitations of analysis clearly described?

-Do the authors discuss how these data can be helpful to advance our understanding of the topic under study?

-Is public health relevance addressed?

Reviewer #1: The conclusions made by the authors are correct in stating that the national model performed better than the local model. However, this analysis doesn't tell us whether RFs are better than ARIMA or a naive baseline.

The authors discuss several legitimate limitations of the study. Another limitation is that they only forecasted 2018 and that there is considerable variation between years 2014-2016 each appear to have had more incidence while 2017 had less. Whether the results are unique to 2018 or generalizable to other years remains to be seen. Also, the model doesn't account for the changes in population susceptibility due to the complex immunological dynamics of dengue (long-term immunity to infecting serotypes and short-term immunity to non-infecting serotypes).

The public health relevance is not addressed.

Reviewer #2: Some statements seem to suggest a causal interpretation that are not warranted. Again, as noted above, there probably needs to be evaluation on more data; I expect the current metrics on the current data to be fairly noisy.

Reviewer #3: The limitations such as the lack of entomological data are addressed. It would be helpful to argue why existing Aedes data was not used, was the required spatial resolution not available?

The public health implications are not sufficiently discussed.

**Editorial and Data Presentation Modifications?**

Reviewer #1: (No Response)

Reviewer #2: (No Response)

Reviewer #3: Here some methodological suggestions for minor revision:

Bagging for decision tree learning hinges on the assumption of independence between observations. The dependence structures between regions for particular feature, but also between different features could be addressed by performing proper outer cross-validation for the random forest.

It might be helpful to compare random forest algorithms with more similar methods such as gradient descent boosting (e.g. xgboost) where carefully tuned models can potentially yield predictors with lower variance. I am not sure whether neural networks is the best suited for methodological comparison, given the limited number of features, the inherent overfitting and lack of accuracy is almost expected.

The authors show validation on the most recent part of the Dengue case count data. To confirm accuracy and robustness, it could be helpful to validate the algorithm also on other parts of the time series (e.g. years 2015-2016).

**Summary and General Comments**

Reviewer #1: This is a quality paper that needs to make one simple but substantial change. The authors have collected a large amount of data across several sources and fit random forest and artificial neural networks to predict dengue incidence in Colombia. Showing that random forests make accurate forecasts of dengue incidence would be a significant addition to the growing literature of using machine learning techniques for disease prediction.

However, the authors only compare these methods to each other and not to a simple baseline method, such as an AR-1 or an ARIMA model, which are likely to be the existing methods of choice by local public health officials. Showing how much machine learning techniques improve over the traditional standards would make for a more compelling argument.

Making the code and data (which are freely available) easily accessible (e.g. in a GitHub repo) would be appreciated.

Reviewer #2: This work compares the predictive performance of models predicting dengue incidence in a given department using (a) an stratified RF model using training instances from the same department only, (b) an RF model using training instances from all departments, and (c) an one-hidden-layer ANN approach, using several types of covariates; it also describes methods for preparing national-level predictions. It demonstrates that, in certain settings and for certain models, prediction accuracy can be improved by the training instance pooling in (b), using the RF-based models presented rather than the ANN approach presented, and that incorporating socioeconomic factors can improve predictive performance.

Major issues:

- Evaluation takes place on a single season. Even with 30 departments, this is probably not reliable. The authors describe the large differences in the different seasons from each other; it is quite reasonable to expect that relative performance of the methods could differ as well. Suggest leave-one-season-out cross-validation or a similar technique.

- The performance of the methods should be compared to existing ARIMA approaches or very simple baselines, such as last-observation-carried-forward.

Comments (mixing technical suggestions that may not be necessary but would

probably strengthen the work's contribution, alongside minor typesetting notes

and more important clarity issues):

- Why is "ANN" a "comparator"? Is it viewed as being worse a priori in this context? Is there literature to back this up? If so, does the architecture of the ANN and the amount of data in this context resemble those based on this literature? Is this applying some pre-existing approach, or not?

- In abstract, several times throughout paper, referring to a "national" model seems a bit confusing or not very descriptive. Suggest finding an alternative description.

- "Furthermore, sociodemographic predictors are important to include to capture longer-term trends in dengue." --- This wording suggests almost a causal interpretation, but the performance analysis is not done using causal inference techniques; it would help to clarify here and any other places that suggest a causal conclusion.

- Line 91: missing "and"

- Line 95: suggests comparison to ARIMA

- Line 106: parametric assumptions may be difficult to test, but I would expect that the untestable assumptions are shared by the nonparametric methods

- Line 107--108: match singular/plural

- Line 109: wording may make it sound like RF is not an ML method

- Line 113: missing comma

- Line 113--115: suggests analyzing importance of all types of variables --- performance impact of covariates / types of covariates in addition to importance rank of individual covariates

- Line 119: Lowe et al. spatiotemporal dengue models in Brazil may qualify, but use a quite different approach

- Line 136: What is the data reporting lag for probable cases? For confirmed cases? Are data revised over time? (At the end of the time series plot, is the drop down from incomplete counts to be revised?) These are important questions and the answers will almost surely indicate that performance metrics for given forecast horizons will need to be approached with some care.

- Line 140: -> "within 14 days"?

- Line 190: Nonparametric models that are not properly regularized will overfit; probably even the very simplest parametric models as well (for the stratified RF model) given the number of covariates.

- Line 213: is there a way for RF to quantify uncertainty, and to evaluate these uncertainty estimates?

- Line 222: What about other hyperparameters, in particular: regularization parameter or number of iterations & learning rate, weight initialization, etc.? in [55], several aspects of the model are adjusted; does this work mirror that one, tune only the # of neurons, or something else? What data is used to evaluate different configurations during the tuning process?

- Line 231: is this performed using an ablation approach or a Shapley value approach? I would expect Shapley value approaches to give a better idea of contributions when certain covariates are redundant or nearly redundant with some other(s), but taken together with these redundant variables are large contributors.

- Line 236: emphasizes why a leave-one-season-out or similar type of evaluation is important.

- Line 253: is there an explanation for the national/pooled RF model's better performance? Is this truly "successful transfer" or is it preventing overfitting?

- Line 254: can MAE and RMSE be put on a more interpretable scale? (Like the relative scale used for feature importance. But some similar options should be avoided if the scaling would directly emphasize instances where one particular method did well or poorly, e.g., by scaling individual errors by the error of one particular system.)

- Line 329--330: RF can also overfit, though.

- Table 3: suggests quantifying the contribution of the AR, Year&Week, weather&vegetation, and sociodemographic categories in terms of error. (But again, care must be taken not to hint without discussion any causal conclusions; top contributors may be confounded by any number of unobserved/unincorporated quantities.)

- Line 322: -> "humans"

- Line 324: shallow ANN's don't have this issue, and there are strategies to avoid these issues for deeper networks

- Line 331: how are the hyperparameters set for RF? If they are tuned based on performance metrics, what training and test data is used to provide these evaluations during the tuning process? What is the exact algorithm for the RF models? Is it really Breiman's original RF or some variant?

- Line 333: this doesn't actually seem like a limitation; probably any model is going to end up with a similar phenomenon. This does heavily suggest comparisons against last-observation-carried-forward, 1-lag or limited-lag linear autoregression, and linear autoregression augmented with week-of-season and year indicator covariates

- Line 339: this seems to be more a problem with the dimensionality than RF; we can plot the point predictions against one or two variables at a time, but issue is that it is a slice/aggregation over values for the other dimensions

- Line 344: this suggest modeling changes in counts or relative changes in counts rather than the counts themselves as a target, perhaps using lagged versions as covariates.

Reviewer #3: The present study suggests the use of a broad range of environmental, climate and socio-economic data on a subnational level to accurately predict Dengue cases in Colombia. While it uses standard methods (random forests) , it would be helpful to address dependence structure in the data more carefully. 

From a practical point of view, the absence of mosquito prevalence data for such predictive tasks and the lack of public health relevance should be addressed more explicitly.

PLOS authors have the option to publish the peer review history of their article (what does this mean?). If published, this will include your full peer review and any attached files.

Reviewer #1: Yes: Stephen A Lauer

Reviewer #2: No

Reviewer #3: No
---

## [Decision Letter · Decision Letter 1]

10 Jul 2020

Dear Prof Zinszer,

Thank you very much for submitting your manuscript "Machine learning and dengue forecasting: Comparing random forests and artificial neural networks for predicting dengue burdens at the national sub-national scale in Colombia" for consideration at PLOS Neglected Tropical Diseases. As with all papers reviewed by the journal, your manuscript was reviewed by members of the editorial board and by several independent reviewers. The reviewers appreciated the attention to an important topic. Based on the reviews, we are likely to accept this manuscript for publication, providing that you modify the manuscript according to the review recommendations. 

The two reviewers and I found the revised version greatly improved, with all major comments from reviewers 1 and 2 satisfactorily addressed. Reviewer 4 is a new reviewer and is asking for more discussion of the pooled vs individual forecast, which I agree with. Please also address all the minor comments made by both reviewers.

Sincerely,

Marc Choisy

Guest Editor

Robert Reiner

Deputy Editor

The authors made a great job in addressing the reviewers' comments. Reviewer 4 is a new reviewer and is asking for more discussion of the pooled vs individual forecast, which I agree with. Please also address all the minor comments made by both reviewers.

Reviewer's Responses to Questions

**Key Review Criteria Required for Acceptance?**

**Methods**

-Are the objectives of the study clearly articulated with a clear testable hypothesis stated?

-Is the study design appropriate to address the stated objectives?

-Is the population clearly described and appropriate for the hypothesis being tested?

-Is the sample size sufficient to ensure adequate power to address the hypothesis being tested?

-Were correct statistical analysis used to support conclusions?

-Are there concerns about ethical or regulatory requirements being met?

Reviewer #2: Methods are now more precisely specified.

Reviewer #4: Please see review.

**Results**

-Does the analysis presented match the analysis plan?

-Are the results clearly and completely presented?

-Are the figures (Tables, Images) of sufficient quality for clarity?

Reviewer #2: Yes, and results are now more complete.

Reviewer #4: Please see review.

**Conclusions**

-Are the conclusions supported by the data presented?

-Are the limitations of analysis clearly described?

-Do the authors discuss how these data can be helpful to advance our understanding of the topic under study?

-Is public health relevance addressed?

Reviewer #2: Yes.

Reviewer #4: Please see review.

**Editorial and Data Presentation Modifications?**

Reviewer #2: (No Response)

Reviewer #4: (No Response)

**Summary and General Comments**

Reviewer #2: The authors have made a thorough revision, addressing my primary concern regarding reliability of the evaluation metrics by adding a cross validation analysis. They have also better contextualized the results by providing an ARIMA baseline for comparison, describing the methods in more detail, and using a more featureful ANN model. My only remaining concern is that some of the text in the response regarding potential revisions to data should be included in the text.

Line 127 -- Line 133: What type of cases are being predicted? Suspected, confirmed, or the sum the two?

Line 128: "confirmation" -> "confirmed cases".

Line 132: Some readers may appreciate the additional information included in the authors' responses regarding the timeliness and nature of revisions, and the reasoning for why it may not impact predictive performance that much. It seems more common that count surveillance systems will undergo large, biased revisions, potentially over longer time periods.

Line 171: it may help to reference the discussion on line 132 here as well, or to rearrange content so that these sections are even closer to each other.

Line 173: "Exponential smoothing approach..." is a sentence fragment. Please specify the decay factor.

Line 222 and elsewhere: "ARMIA" -> "ARIMA"

Line 338: (Again, I wouldn't see this as a limitation per se; it is likely a feature of many of the best existing predictors for various epidemiological forecasting tasks.)

Line 173: Please specify the decay factor for the exponential smoother.

Line 173: "Exponential [...] data to estimate" -> "An exponential [...] data was used to estimate"

Line 176: "local level" -> "the local level"

Line 198: "forest that is" -> "forecast, which is"

Line 204: "hidden layer" -> "hidden layers"

Line 244: "therefore not suitable to conduct" -> "therefore was not suitable for conducting"

Line 338: (Again, I don't see this as a limitation; it is likely a feature of many of the best predictors for various epidemiological forecasting tasks.)

Reviewer #4: The manuscript was well written and clear for the most part. The only pieces missing are clarification on a few points made.

PLOS authors have the option to publish the peer review history of their article (what does this mean?). If published, this will include your full peer review and any attached files.

Reviewer #2: No

Reviewer #4: No
---

## [Editor Report · Decision Letter 2]

12 Aug 2020

Dear Dr Zinszer,

We are pleased to inform you that your manuscript 'Machine learning and dengue forecasting: Comparing random forests and artificial neural networks for predicting dengue burdens at the national sub-national scale in Colombia' has been provisionally accepted for publication in PLOS Neglected Tropical Diseases.

Best regards,

Marc Choisy

Guest Editor

Robert Reiner

Deputy Editor

---

## [Editor Report · Acceptance letter]

17 Sep 2020

Dear Dr Zinszer,

We are delighted to inform you that your manuscript, "Machine learning and dengue forecasting: Comparing random forests and artificial neural networks for predicting dengue burdens at the national sub-national scale in Colombia," has been formally accepted for publication in PLOS Neglected Tropical Diseases.

Best regards,

Shaden Kamhawi

co-Editor-in-Chief

Paul Brindley

co-Editor-in-Chief
